# Effects on Spirulina Supplementation on Immune Cells’ Parameters of Elite College Athletes

**DOI:** 10.3390/nu14204346

**Published:** 2022-10-17

**Authors:** Yuting Zhang, Yan Zhang, Wei Wu, Yining Xu, Xiaohan Li, Qiner Qiu, Haimin Chen

**Affiliations:** 1Faculty of Sports Science, Research Academy of Grand Health, Ningbo University, Ningbo 315211, China; 2Collaborative Innovation Center for Zhejiang Marine High-efficiency and Healthy Aquaculture, Ningbo University, Ningbo 315211, China; 3State Key Laboratory for Managing Biotic and Chemical Threats to the Quality and Safety of Agro-Products, Ningbo University, Ningbo 315211, China

**Keywords:** spirulina, immune cell, athlete, training

## Abstract

Objective: To identify the effect of spirulina supplementation on the immune cells’ indicators of young soccer players during the preparation period of a tournament. Methods: 39 undergraduate male soccer players were recruited and randomly allocated into a spirulina supplementation group (SP group, *n* = 20) and the placebo supplementation group (PB group, *n* = 19). Their elbow venous blood samples were collected before and after the preparation period of a tournament, which included 8 weeks total. The differences within the group and between groups were recorded and analyzed. Results: The ratio of the basophils in the SP group between the pre-test and post-test were statistically significantly different (*p* < 0.05). In the PB group, the percentage of before and after in leukocytes and monocytes were statistically significantly different (*p* < 0.05). In the data of the post-test, the percentage of monocytes and basophils between the SP group and PB group were statistically significantly different. The delta variations of monocytes between groups were significantly different (*p* < 0.05). Conclusions: Intense long-duration exercise can reduce the ratio of leukocytes and monocytes in young athletes, yet the spirulina supplement can inhibit the change. It also might improve immunity to parasites, pathogenic bacterium, and rapid-onset allergies.

## 1. Introduction

Immune cells play different roles so as to maintain immune function. For instance, the functions of granulocytes (Neutrophil, eosinophil, basophil) mainly exist for phagocytose foreign substances and for participating in allergic reactions [1]. Moreover, the monocytes can differentiate the macrophages and dendritic cells to be involved in the function of the immune system. These parameters are widely used to estimate the immunity of athletes [2]. High-intensity or prolonged periodic exercise often suppresses athletes’ immune function [3,4]. The athletes are often exposed to a training environment with high intensity and a volume load which may cause their immune system to experience a stress response, becoming weaker than normal and inducing a negative effect on health [5,6]. Generally, abnormal changes in the ratio of immune cells are often regarded as a representation that is symptomatic of immune dysfunction [7]. Many athletes are plagued by similar problems. Fluctuations in immunity can affect not only training but also competitive performance during important competitions [8]. Therefore, it is extremely important for athletes to ensure the normalization of immune levels after daily training [9].

Currently, athletes usually take supplementations to maintain health and speed up the recovery, including immunity after training [10]. However, athletes are very careful when choosing sports supplements due to the strict requirements in matches. The production of certain dietary supplements may inadvertently produce substances that are prohibited in competition [11]. Besides, regarding various supplements currently on the market, such as creatine, the edible safety of long-term use has not been studied. For the avoidance of violations in doping tests, in recent years, purely natural supplements have attracted more attention from coaches, physical strength trainers, and sports dieticians [12]. Natural supplements, which are defined as those obtained from naturally grown plants or animals by non-chemical synthesis methods, are considered to have higher bioavailability and better efficacy than extracts or synthetic supplements [13].

Spirulina, which is a natural supplement rich in a variety of substances that have been proven to regulate immune function, are gaining popularity and can be prescribed by physicians and pharmacists to aid in clinical treatment [14]. Gamma-linolenic acid, one of the substances in spirulina, has been proven to modulate the immune system [15]. An animal study showed that gamma-linolenic acid can improve rats’ immunity [16], and, similarly, researchers also discovered that it regulates the immune response in cell models [17]. Also, another substance in spirulina called phycocyanin was certified for its regulation effect on the immune system [18]. The function of immune modulation in animal experiments has also been certified, and the researchers believed that its antioxidative and oxygen-free radical scavenging properties may contribute, at least in part, to its modulation of the immune system [19]. Nevertheless, results regarding humans are few. The functions of antioxidants in spirulina were certified by some researchers [20]. For studies on immune function regarding spirulina intervention, there are few studies that use immune cells as evaluation indicators in particular.

In this study, we used a randomized, double-blind experiment to conduct a spirulina and placebo intervention experiment on 39 college soccer players who participated in the training of the 16th Universities’ Games in Zhejiang Province soccer league to explore the effect of spirulina on the proportion of leukocytes, basophiles and monocytes in the peripheral blood of high-level athletes.

## 2. Materials and Methods

### 2.1. Participant Recruitment and Experimental Ethics

Recruitment was conducted by issuing directional network questionnaires to Ningbo university’s high-level sports team from 26 January 2022 to 26 February 2022. The inclusion criteria of the trials were as follows: (1) Be at least national secondary athletes (with the sports level certificate issued by the General Administration of Sport of China); (2) be without any record of sports injury, infectious disease, or immune system disease clinically diagnosed within one month; (3) be without any hereditary blood diseases; (4) be without any history of smoking, cardiovascular, or severe respiratory disorders; (5) did not have a cold or uncomfortable symptoms in the past month; (6) had no intake of drugs or supplements in the past month. The follow-up exclusion criteria to determine whether participants could continue the trial were as follows: (1) Suspension of training due to sports injury during the experiment; (2) someone who has been absent more than one-fifth of the total of training; (3) someone infected with a cold, feeling unwell or taking drugs or other supplements; (4) the number of times absent exceeds the prescribed number of one-fifth. The participant’s data was excluded if any one of the above was fulfilled. The process of screening was conducted on 27 February 2022.

Eventually, after the screening for the inclusion criteria, 39 participants were recruited in this trial with their basic information provided in Table 1. All participants had consented before signing the informed consent and they were informed of the risks and benefits of the study. The procedures obey the Declaration of Helsinki. The Ethics Committee of Ningbo University has also reviewed this project. Ethical code: ChinCTR2100045524.

According to the baseline, there was no statistically significant difference between the two groups at baseline. The scheme of the trial procedure was presented in Figure 1.

### 2.2. Design

This was a randomized double-blind placebo-controlled trial. All participants were numbered 1–39 according to the order of registration. One of the researchers used the SPSS (version 24.0, IBM, Armonk, NY, USA) random number generator to divide the groups, allocating all eligible participants into the spirulina supplementation group (SP) (*n* = 20) and the placebo supplementation group (PB) (*n* = 19). Another researcher who was not aware of the allocation and the tablets gave spirulina and placebo to participants. Two groups of researchers were responsible for one group, respectively. A day’s dose was issued to athletes after the completion of daily training (6p). An independent researcher made a record of the participant’s intake according to the regulations by asking.

The spirulina used in this trial was certified to be safe (the spirulina tablets were provided by Lijiang Chenghai Bao Biological Development Co., Ltd., Lijiang, China). The approval number was G20120502. The execution standard was Q/LBE 0002 S-2014. Each tablet contains 500 mg of spirulina extract. The detail of the trial conduction was provided in Table 2. The placebo was a hollow capsule made of edible starch and had the same color as spirulina tablets. The method of intake was the same as spirulina.

Each participant was given a daily intake of 3 g (Juszkiewicz,2020) supplementation or placebo. The supplementation was given at 8:00 AM, 2:00 PM, and 8:00 PM, with 2 tablets each time and 6 tablets total each day.

### 2.3. Design Sampling and Detection

The formal trial started on 1 March 2022, which was also the day of first training after the participants’ enrollment. The first blood sampling was taken on 11 April 2022; at the time, that was 24 h before the training. After the first blood sampling, 8 weeks of training was conducted from 11 April 2022, to 6 June 2022. The second blood collection was taken on 7 June 2022; at the time, that was 24 h after the last training. The timeline of the trial is presented in Figure 2.

Since the leukocytes and its subsets would stabilize at 12 to 24 h after exercise stimulation [21,22,23], the first blood sampling, in which the blood of the elbow venous was collected, was set to be 24 h before the preparation training, while the second was set to be 24 h at the end of training. The whole interval between two blood samplings was 8 weeks. The researchers notified the participants 48 h before each blood sampling to make sure that they would not intake any drink or food that could influence the blood indicators. The sampling and blood cell testing were carried out by doctors from the blood laboratory department of Ningbo Hospital.

### 2.4. Training Plan

The participants took part in a training plan of 150 min per normal training day. They rested every Sunday. The training was conducted on the training grounds of the campus. The whole plan is presented in Table 3. The coach would attempt to make the athlete complete the daily training plan. The coaches used heart bands (Garmin, HRM-Pro Plus, China) to monitor the intensity of the athletes. Unlike other training programs, we used the relative intensity to train the athletes, gradually increasing the absolute intensity due to the adaptability of the individual to the training.

### 2.5. Statistical Analysis

The data analysis software used for statistical analysis was SPSS (version 24.0, IBM, Armonk, NY, USA). The data was expressed in the form of “standard ± deviation”. The statistically significant level was set at 0.05. First, normal distribution of the study variables was verified with a Shapiro-Wilk test. If the result showed that the data in this study are normally distributed, the differences within each group between the baseline and endpoint would be analyzed by using the paired *t*-test. Moreover, the difference of the effect between groups and the delta variation between before and after would be assessed by the independent samples *t*-test. If the test data do not fit a normal distribution, then a nonparametric test and Mann-Whitney U test would be used in analyzing the data.

## 3. Results

### 3.1. The Parameters of Immune Cells before and after the Supplementation

The participants completed the eight-week trial, and no participants were excluded based on follow-up exclusion criteria. The results were all within the clinical reference range [24]: Leukocyte (10^9^/L): 4–10, Neutrophil%: 50–70, Lymphocyte%: 20–50, Monocyte%: 3–8, Eosinophils%: 0.5–5, Basophilic%: 0–1.

As was shown in Table 4, after the 8 weeks of intense training, the placebo intaking group’s leukocytes reduced significantly (*p* < 0.05), and their ratios of monocytes reduced as well. (*p* < 0.05). Yet, the ratios of basophils in the group upon spirulina supplement increased (*p* < 0.05). In addition to this change, the other parameters in the SP group were insignificant during the training.

### 3.2. Difference of the Post-Test between the SP Group and PB Group

By comparing the data of the post-test between the two groups, the participants’ ratios of monocytes upon spirulina supplement were higher than the placebo group (*p* < 0.05) and their ratios of basophils were higher (*p* < 0.05) on average. This is shown in Figure 3.

### 3.3. Comparison of the Delta Variation before and after between SP Group and PB Group

The result in Table 5 shows that the delta variation of the monocytes ratio between groups is statistically significantly different (*p* < 0.05). The PB group became lower than before. * *p* < 0.05.

## 4. Discussion

The purpose of this study was to identify if the spirulina supplement inhibits the ratios of immune cells changing during a tournament in young athletes. In this study, we chose the leukocyte and its subtypes as our observation indicators. Leukocytes are commonly used as an assessment tool to help team doctors and coaches to understand the health condition of players [25]. Leukocyte cells can be divided into five subtypes: monocyte, eosinophils, neutrophils, basophils, and lymphocytes. They are the basic functional units of the immune system. Leukocytes fight infections and foreign organisms in the body. Furthermore, their sub-types play different roles in immune function. Lymphocytes mainly identify antigens. The functions of granulocytes (Neutrophil, eosinophil, basophil) are mainly for phagocytose foreign substances and participate in allergic reactions. The monocytes can differentiate the macrophages and dendritic cells to participate in immune function. The immune cells maintaining themselves in a clinical reference range means the immune system is in good health.

In the PB group, we found that leukocytes and the ratio of monocytes changed to be significantly lower after intense training. This means that the 8-week training caused the athletes’ immune cells to change. Published evidence has shown that long-duration or intense sports training can weaken the immune system [3]; for instance, the significant decrease of the ratio of immune cells. When the data were compared between the two groups, the drop was more pronounced in the placebo group. A similar phenomenon has been verified by previous studies in which vigorous exercise significantly reduces the ratio of immune cells in athletes [25]. Horn et al. have proved that prolonged intense exercise can weaken humans’ immune function [26,27], and this includes a decrease in the rate of immune cells. Daniela et al. have opined in a study about aging that the ratio of immune cells decreasing means the basis of the immune system weakens [28]. Córdova A et al. verified that intense training could decrease the number of an athletes’ leukocytes [29]. Furthermore, published evidence demonstrated that high intensity training can decrease monocyte function. Yet, more studies conclude that sports intervention can increase the number of monocytes. The increase observed in monocytes can be associated with muscle tissue remodulation due to possible damage caused by training load [30]. The reason they came to different conclusions might be due to the different methods used in their studies. Leukocytes are part of the immune system, helping the body to fight infections and foreign organisms. The declining of leukocytes in the PB group could be a sign of body damage. It could also be due to a decline in immune function. The PB group’s ratio of monocytes was decreased as well. Monocytes gather in the infected tissue within 8 to 12 h in the presence of inflammatory signals, and their function involves differentiating into macrophages and dendritic cells to produce an immune response [31]. Therefore, its decline might be adverse to immunity, even though there is no published evidence proving that the monocytes’ proportion in leukocytes is extremely minimal. In a word, the preparation period of a tournament can decrease the ratio of immune cells in the present trial. There was no statistically significant alteration in the ratio of neutrophils and lymphocytes in the participating athletes in this trial. The phenomenon might indicate that there was an adaptation to training loads in the participants as well.

We found that the leukocytes and monocytes changed to be lower than pre-test in the athletes of the PB group after the 8-week intense training. Nevertheless, it was found in the SP group that the spirulina supplementation can inhibit this change in immune cells, which may be helpful in improving athletes’ immunity and resistance to pathogen infection. Thus, it is assumed in this study that the spirulina supplementation could regulate immune system function in elite college athletes.

Nonetheless, most of the parameters, except the eosinophils of the SP group, had not significantly changed. There were some investigations that had analogous conclusions. Milasius et al. concluded that the percent ratio of agranulocytes and granulocytes leveled off after the supplementation [32]. Juszkiewicz’s team found that the lymphocytes changed insignificantly after the spirulina supplement in a study of elite rowers [33]. The eosinophils’ change in both groups was higher; however, the SP group changed significantly. The phenomenon of an increase in eosinophils has also been found in some other studies. Other parameters changed to be statistically insignificant from baseline to an endpoint, meaning that the participants in the SP group experienced no adverse change during a high intense intervention of 8 weeks. Nevertheless, much published evidence has shown that the immune cells changed after high intense training and that most conclusions showed that the parameters changed to be lower than before after 24 h. Therefore, the spirulina supplement might improve the phenomenon of immune cells changing due to training, maintaining the immune cells in a normal range.

The difference in monocytes may be due to its decline of the PB group. Moreover, the supplement might maintain the ratio of the monocytes in the SP group. Grenon et al. found that gamma-linolenic acid, one substance in spirulina, can alter the functions of monocytes [34]. So, the ratio of monocytes may be changed with the function changing. Many researchers checked that the basophils would change to be lower than before training. There was an investigation demonstrating that the training cannot influence the number of basophils [30], but it intended to verify the cell ratio. The heterogeneity may be due to the different design, and even the different sports. In our trial, the basophils in the SP group changed insignificantly, but were significantly higher than in the PB group. Therefore, the supplement might maintain this parameter in a steady range.

There is a limitation in this trial where due to the small sample number of female soccer players, they were not studied herein. When it comes to the gender subgroups, the immune responses may be affected by the gender factor. Fragala’s team (2011) considered that some immune parameters in different genders were significantly different after exercise in animals [35]. Many studies have pointed out that estrogen and testosterone both are essential factors in the immune response. The female immune system is dominated by estrogen regulation. In contrast, the male immune system is dominated by testosterone. For example, an investigation attributed sex differences in B lymphocyte immune responses to the protective responses of estrogen. A study conducted by Kraemer’s team in 1998 demonstrated this [36]. Furthermore, a cycling endurance study verified the differences in the response of lymphocytes in different genders [37]. Gender differences in basophils’ ratio changes after spirulina supplementation require further study. The mechanism still needs to be further explored.

## 5. Conclusions

Eight weeks of intense training significantly decreased the athletes’ leukocytes and their ratios of monocytes. However, the spirulina supplement group did not experience this phenomenon. So, we assumed that spirulina can indeed inhibit the change in immune cells and strengthen the eosinophils and basophils. Therefore, the spirulina supplement was beneficial to the stabilization of the ratio of leukocytes, monocytes, eosinophils, and basophils in elite university athletes. Furthermore, spirulina has a positive effect on rapid-onset allergies and responses to parasites or pathogenic bacterium, aiding in the improvement of immune regulation.

## Figures and Tables

**Figure 1 nutrients-14-04346-f001:**
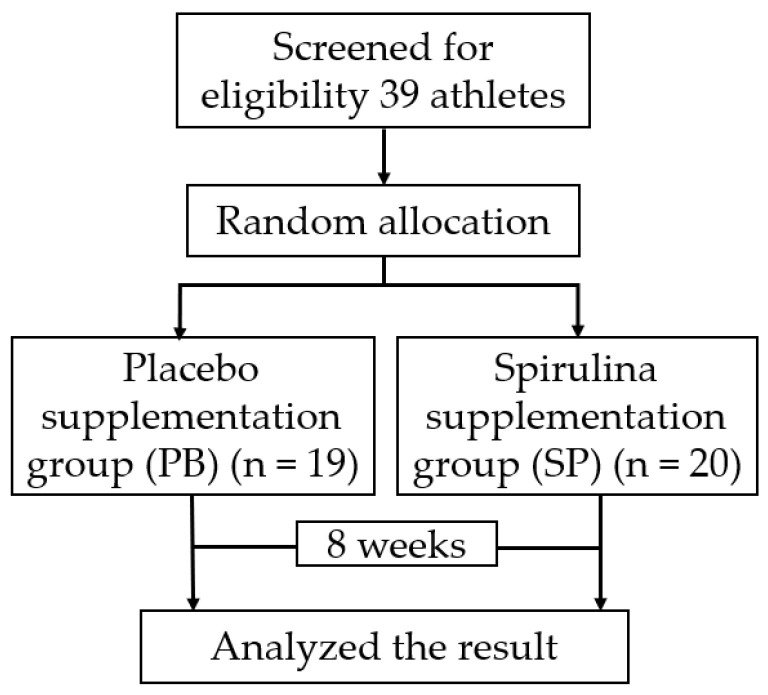
An experimental flow graph; the participants’ grouping and flow of procedure.

**Figure 2 nutrients-14-04346-f002:**
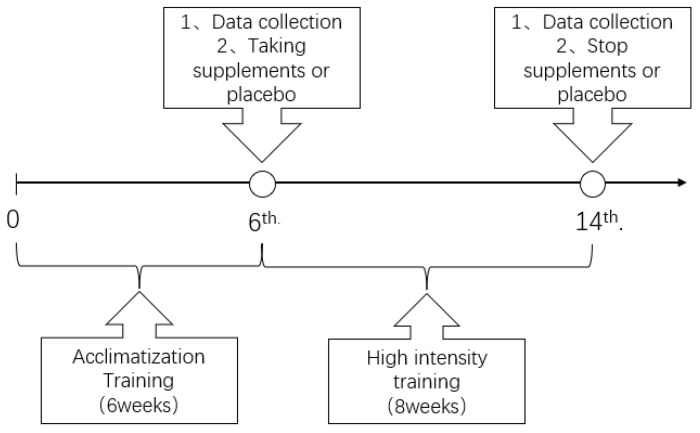
The timeline of the trial.

**Figure 3 nutrients-14-04346-f003:**
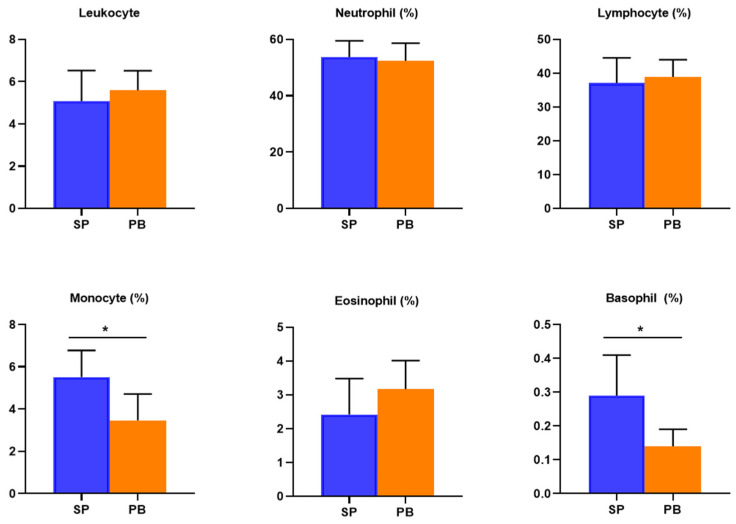
The difference of the post-test between the groups. Notes: Except the unit of leukocytes being 10^9^/L, the other parameters’ units are %. * *p* < 0.05.

**Table 1 nutrients-14-04346-t001:** The characteristics of the participants in baseline.

Variables	PB Group	SP Group	*t*	*P*
Sex	Male	Male	/	/
Number	19	20	/	/
Age (year)	19.69 ± 0.92	19.87 ± 0.74	0.28	0.78
Body height (cm)	177.72 ± 5.34	179.86 ± 2.70	1.91	0.07
Body mass (kg)	62.36 ± 9.20	64.95 ± 7.792	−0.49	0.62

Notes: Mean ± standard deviation (SD) unless otherwise specified.

**Table 2 nutrients-14-04346-t002:** The ingredient list of spirulina and placebo (per 100 g).

Supplementation	Component	Content	Component	Content
Spirulina tablets	Protein	60–75 g	Zn	0.8–3 mg
	Carbohydrate	10–30 g	Se	1–3 μg
	Fat	2–3 g	β carotene	50–200 mg
	Dietary fiber	0.5–1.5 g	Vitamin B1	100–300 μg
	γ-linolenic acid	0.5–1.6 g	Vitamin B2	3–10 μg
	Spirulina polysaccharide	2.6 g	Vitamin B6	100–300 μg
	Docosahexaenoic acid	0.2–0.8 mg	Vitamin B12	10–20 μg
	Coenzyme Q10	1.5–4 mg	Vitamin C	10–80 μg
	Chlorophyll	0.8–1.2 g	Vitamin E	1–8 μg
	Carotenoid	100–400 mg	Nicotinic acid	2.5–10 μg
	Phycocyanin	3.5–7 g	Folic acid	35–100 μg
	Ca	40–150 mg	Sour regurgitation	0.5–1.5 μg
	Fe	20–50 mg	Inositol	≥2 μg
Placebo capsules	Edible starch	100 g		

**Table 3 nutrients-14-04346-t003:** Details of the training plan in the trial.

Phase	Exercise Items	Frequency	Description
**Warm up**	Jogging	Every training day	Start from jogging, the speed which induced the HR of participants increases to 60% HRmax.
	Whole-body Dynamic Stretch, contain crawling, lunge, and shoulder and hip mobility exercises.	/	A 20-min whole body dynamic stretch exercise, the HR in the dynamic stretch will be maintained from 50–60% HRmax, and the time in which the HR less than 50% HRmax should be no longer than 2 min.
**Workout**	Skill	Every training day	Start from progressing with the ball; the speed which induced the HR of participants increased to 60% HRmax.
	Personal skill training, passing and catching drills.	/	A 15-min Personal skill training, a 15-min passing and catching drills training and a 15-min team cooperation training. The HR during the training will be maintained from 60–70% HRmax and the time in which the HR less than 50% HRmax should be no longer than 5 min.
**Competition time**	Match and instruct.	3 times a week	Starting from 3v3, the training which induced the HR of participants increases to 90% HRmax.
	3–11/3–11 training, cooperation training, teamwork training.	/	After a short rest to 50%HRmax, start a 5v5/8v8/11 match or friendly match. The coach will instruct tactics.
**Strength/endurance training**	Jumping or running.	Once a week	Start from a squat jump/shuttle run, the training which induced the HR of participants increases to >90% HRmax
	Squat jump, lunge jump, tuck jump. Varied pace running, shuttle run.	/	After a short rest to 60%HRmax, perform training lunges; jump and tuck; jump/varied pace running. The time between sets less than 3 min, 90% HRmax.
**Cool-down**	Stretching/jogging.	Every training day	A 20–30 min cooldown to close at resting heart rate (RHR).
	Static stretching, respiratory training.	/	
**Total time**	150 min		

**Table 4 nutrients-14-04346-t004:** A comparison of immune cells ratios within groups.

Group	Item	Leukocyte	Neutrophil	Lymphocyte	Monocyte	Eosinophil	Basophil
**PB**	before	6.71 ± 1.56	52.44 ± 8.41	37.89 ± 5.65	5.99 ± 0.91	2.30 ± 1.26	0.28 ± 0.17
	after	5.60 ± 0.92	52.44 ± 6.24	38.83 ± 5.50	3.47 ± 1.24	3.06 ± 0.83	0.15 ± 0.05
	*p*	0.04 *	0.99	0.80	0.01 *	0.18	0.12
**SP**	before	6.05 ± 1.09	51.97 ± 7.91	37.36 ± 10.35	5.79 ± 1.12	1.61 ± 0.76	0.29 ± 0.12
	after	5.42 ± 0.98	54.39 ± 5.59	37.05 ± 7.46	5.52 ± 1.26	2.41 ± 1.06	0.25 ± 0.24
	*p*	0.14	0.23	0.92	0.59	0.01 *	0.55

Notes: Before means the first-time collection, after means the second time collection. Except where the unit of leukocytes is 10^9^/L, the other parameters’ units are %, * *p* < 0.05.

**Table 5 nutrients-14-04346-t005:** A comparison of immune cells ratios within groups.

Variables/ Group	SP Group (*n* = 20)	PB Group (*n* = 19)	*p*
Leukocyte	−0.63 ± 1.45	−1.11 ± 1.24	0.45
Neutrophil	2.41 ± 6.26	−0.01 ± 12.72	0.58
Lymphocyte	−0.31 ± 10.96	0.94 ± 9.30	0.80
Monocyte	−0.27 ± 1.74	−2.51 ± 1.37	0.01 *
Eosinophil	0.88 ± 0.80	0.76 ± 1.43	0.84
Basophil	−0.05 ± 0.25	−0.13 ± 0.18	0.45

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
