# Peer review of "Effects on Spirulina Supplementation on Immune Cells’ Parameters of Elite College Athletes"

_nutrients, 2022, doi:10.3390/nu14204346_

Round 1

Reviewer 1 Report

Dear Authors, 

the manuscript entitled "Spirulina supplement inhibits the imbalance of immune cells in young athletes after intense training" aims to evaluate the effectiveness of spirulina on the immune system of athletes during strenuous training phases.

There are major critical issues to overcome, especially relating to the introduction section, the description of the subjects, the training program and statistical analysis.

INTRODUCTION:

the introduction section is not oriented to the aim of the study. It appear too general and the reference related to athletes appear few. 

The reviewer advises to reorganize the chapter, orienting it more towards sports subjects and the purpose of the study.

- line 36-38: please provide a reference;

- line 39-42: please provide a reference related to athletes subjects;

- line 47-49: please provide a reference related to athletes subjects;

- line 63-75: there is no reference and in the reviewers' point of view some aspects should be placed before in introdiuction and other in discussion section.

- the period from line 76 to 81 should be removed from introduction section.

METHODS

The chapter contains few details on the subjects enrolled (sex, sport practiced, etc.), on the type of training and on the composition of the placebo.

- table 1: what was the sex of the sample?

- what was the sport practised?

- what was the meaning of tournament? Different sport have different peridization with different training strategies.

- the Follow-up exclusion criteria does not appear useful as 2.5 section. Should be placed in 2.1 section. 

- it is not clear what placebo was made of;

- table 3: in day 4 the last 2 rows not appear correct;

- the training program is not well describe: all athletes performed the same program also from different sport and with different sex? What was the intensity of the training?

- the statistical analisys appear too poor, the analisys of delta variation (post-pre) should be more appropriate in order to evaluate the magnitude of effect in addition to the ratio.

- the sentence at line 159: "A nested design was used to compare the difference between the intake group and the blank group" should be rewritten.

RESULTS:

- figure 2 and "3.1 Procedure" should be placed in methods section;

- line 167-169: what was the other parameters? For example, what was the weight of the two groups?

- "3.2 The spirulina supplement can inhibit the decline of leukocytes and monocytes’ ratio due to the intense training": the intensity of the training was not describe before.

- What was the " steady range" describe at line 69?

- what was the differences at baseline between groups?

- the procedure describe at line 156 "The subgroup analysis would be made to assess the gender difference by applying another paired t test" not appear in results section.

- the procedure that figure 3 describe is not presented before in methods section. 

Author Response

Dear Reviewer,

    Thank you very much for your attention and comments on our manuscript entitled “Spirulina supplement inhibits the imbalance of immune cells in young athletes after intense training”. Those professional comments and advice are all valuable and assuredly helpful for our revising and improving to our manuscript, as well as the vital guiding to our future research. We have made corrections accordingly which we expect to meet with your approval. Revised portions are highlighted in red. We also revised according to two reviewers. We look forward to hearing from you.

Reviewer 2 Report

The manuscript is well written and provides new insight into spirulina supplementation. 

L76-81 Authors should write the aim of the study rather than the findings. I suggest revisiting it and presenting the aim without coming directly to conclusions.

Statistical analysis

Since authors are analyzing the subgroups, a correction method (Bonferroni, BH, etc) should be applied to reduce FWER or FDR

L139 Why the authors did not consider also the changes occurring to the training? 

L182 Authors should avoid writing conclusions in the titles. Just name it "Difference of the post-test between the groups" and remove the rest. (as well as for the 3.2 paragraph)

Author Response

Dear Reviewer,

    Thank you very much for your attention and comments on our manuscript entitled “Spirulina supplement inhibits the imbalance of immune cells in young athletes after intense training”. Those professional comments and advice are all valuable and assuredly helpful for our revising and improving to our manuscript, as well as the vital guiding to our future research. We have made corrections accordingly which we expect to meet with your approval. Revised portions are highlighted in red. We also revised according to two reviewers. We look forward to hearing from you.

Have a good day,

Yuting Zhang.

Round 2

Reviewer 1 Report

Dear authors,

the revision process significantly improved the quality of the manuscript.

Although most of the previous comments have been addressed, in my opinion I believe that the statistical analysis is currently at a basic level and that it should be more in-depth, perhaps relying on a statistician: this aspect would improve the scientificity of the results obtained.

Author Response

Thank you very much for your professional advice on my manuscript. We revised it according to your request. Please see the attachment.

Reviewer 2 Report

The authors correctly addressed my suggestions and, following the Reviewer #1 the authors really improved the manuscript.

Author Response

    Thank you very much for your attention and comments on our manuscript. Those professional comments are all valuable and very helpful for revising and improving our manuscript, as well as the important guiding to our future research. 

Yuting Zhang,

Have a good day!